# The Predictive Value of the Cervical Consistency Index to Predict Spontaneous Preterm Birth in Asymptomatic Twin Pregnancies at the Second-Trimester Ultrasound Scan: A Prospective Cohort Study

**DOI:** 10.3390/jcm9061784

**Published:** 2020-06-08

**Authors:** Johannes van der Merwe, Isabel Couck, Francesca Russo, Xavier P. Burgos-Artizzu, Jan Deprest, Montse Palacio, Liesbeth Lewi

**Affiliations:** 1Department of Development and Regeneration, Cluster Woman and Child, Group Biomedical Sciences, KU Leuven University of Leuven, 3000 Leuven, Belgium; isabel.couck@uzleuven.be (I.C.); francesca.russo@uzleuven.be (F.R.); jan.deprest@uzleuven.be (J.D.); liesbeth.lewi@uzleuven.be (L.L.); 2Division Woman and Child, Department of Obstetrics and Gynaecology, University Hospitals Leuven, 3000 Leuven, Belgium; 3Fetal i + D Fetal Medicine Research Center, BCNatal-Barcelona Center for Maternal-Fetal and Neonatal Medicine, Universitat de Barcelona, 08028 Barcelona, Spain; xpburgos@clinic.cat (X.P.B.-A.); MPALACIO@clinic.cat (M.P.); 4Transmural Biotech S. L. Barcelona, 08028 Barcelona, Spain; 5Institute for Women’s Health, UCL, London WC1E 6HU, UK

**Keywords:** spontaneous preterm birth, twin pregnancy, cervical consistency index, uterocervical angle, cervical texture, cervical length

## Abstract

Novel transvaginal ultrasound (TVU) markers have been proposed to improve spontaneous preterm birth (sPTB) prediction. Preliminary results of the cervical consistency index (CCI), uterocervical angle (UCA), and cervical texture (CTx) have been promising in singletons. However, in twin pregnancies, the results have been inconsistent. In this prospective cohort study of asymptomatic twin pregnancies assessed between 18^+0^–22^+0^ weeks, we evaluated TVU derived cervical length (CL), CCI, UCA, and the CTx to predict sPTB < 34^+0^ weeks. All iatrogenic PTB were excluded. In the final cohort of 63 pregnancies, the sPTB rate < 34^+0^ was 16.3%. The CCI, UCA, and CTx, including the CL was significantly different in the sPTB < 34^+0^ weeks group. The best area under the receiver operating characteristic curve (AUC) for sPTB < 34^+0^ weeks was achieved by the CCI 0.82 (95%CI, 0.72–0.93), followed by the UCA with AUC 0.72 (95%CI, 0.57–0.87). A logistic regression model incorporating parity, chorionicity, CCI, and UCA resulted in an AUC of 0.91 with a sensitivity of 55.3% and specificity of 88.1% for predicting sPTB < 34^+0^. The CCI performed better than other TVU markers to predict sPTB < 34^+0^ in twin gestations, and the best diagnostic accuracy was achieved by a combination of parity, chorionicity, CCI, and UCA.

## 1. Introduction

Preterm birth remains a significant problem in obstetrics responsible for more than 70% of neonatal and infant deaths and affecting about 1 in 10 pregnancies [1,2]. In twin pregnancies, the rate of preterm birth is almost ten times higher than in singletons [3]. Antenatal care programs aspire to prevent spontaneous PTB (sPTB,) and therefore its consequences. The current standard of care for sPTB screening in singletons entails mid-gestational transvaginal ultrasound (TVU) cervical length (CL) measurement, [4] and prevention consists of vaginal progesterone for those at increased risk [5]. In twins, CL has a low sensitivity; hence, women at risk are not being identified [6,7]. Moreover, at present, there is no useful intervention to prevent sPTB in twins, but this could be due to the misidentification of women at risk [8].

Additional TVU markers, including funnelling and sludge, have been evaluated in twins but with limited success [9]. Recently, the cervical consistency index (CCI) outperformed CL in high-risk singleton pregnancies [10]. The CCI measures the maximum deformability of the cervix during a standard TVU examination [11]. Overall, the CCI has a clear inverse linear relationship with gestational age (GA) and is reproducible and effective in the prediction of sPTB [12], but so far, it did not show any clinical utility in a small prospective twin study ((area under the receiver operating characteristic curve (AUC) = 0.488 to predicting sPTB before 34^+0^ weeks) [13].

Another novel ultrasound marker that has been evaluated in singletons is the uterocervical angle (UCA) [14,15]. Once again, initial results indicate that UCA, either alone or combined in a prediction model, can serve as an effective predictor of sPTB in singletons [16]. One retrospective study in twins showed that an UCA > 110° was associated with an odds ratio (OR) 15.7 (7.2–34.4) for sPTB before 32^+0^ weeks [17].

Finally, the quantitative analysis of cervical texture (CTx) of standard TVU cervical grayscale images was independently associated with sPTB before 37^+0^ weeks in singletons [18]. This promising approach is also feasible in twins, albeit with limited success in predicting sPTB [19].

The purpose of this prospective study was to evaluate the novel TVU markers CCI, UCA, and CTx alongside standard CL measurement at mid-pregnancy in asymptomatic twin gestations, to predict sPTB before 34^+0^ weeks.

## 2. Experimental Section

This prospective cohort study recruited women with twin pregnancies who had a TVU assessment between 18^+0^ and 22^+6^ weeks as part of a study on the role of the Arabin pessary in the prevention of sPTB between October 2016 and January 2019. Ultrasound examination was done by ultrasound examiners trained in measuring the CL (CLEAR program from the Perinatal Quality Foundation; https://clear.perinatalquality.org/). Analyses of all images were done off-line. Gestational age was calculated on the basis of the crown-rump length measurement of the larger twin in the first trimester for spontaneous pregnancies. In pregnancies resulting from in vitro fertilization, gestational age was defined using the date of conception in fresh cycles or embryonic age in frozen-thawed cycles. Information on baseline demographic characteristics and obstetric history were prospectively collected. Perinatal outcomes were retrieved from hospital files.

sPTB was defined as spontaneous preterm delivery or induction of labour because of preterm rupture of membranes. Women were excluded from the study if treatment to prevent sPTB (cervical cerclage or cervical pessary) was instituted or if a fetal abnormality was diagnosed on ultrasound. Moreover, women were excluded from the analysis if they delivered preterm because of a medical or fetal indication (e.g., caesarean delivery or induction of labour because of pre-eclampsia, fetal growth restriction), if they were lost to follow-up, or if the ultrasound images did not meet the quality criteria described below. As part of institutional care, the NICE guidelines were followed to start low dose aspirin for pre-eclampsia prevention, and dichorionic pregnancies were advised to deliver by 37^+0^–38^+0^ weeks while monochorionic pregnancies were advised to deliver by 36^+0^–37^+0^ weeks [20]. This study was nested in a larger study, which was approved by the local Ethics Committee Research UZ/KU Leuven (ECD S58820). All participants provided informed consent for TVU.

### Image Acquisition and off-Line Analysis

For TVU image acquisition, a Voluson E8 or E10 (GE Medical Systems, Zipf, Austria) equipped with a vaginal probe with a frequency of 2–10 MHz was used. A set of images was acquired with the woman in lithotomy position after emptying her bladder. Firstly, a sagittal view of the cervix without exerting any pressure with the transducer was obtained, as described previously [21], upon which the cervical canal and the internal and external cervical os were clearly seen. This image was used for the CL and UCA measurement, and to assess for funnelling or sludge presence. A second image was then saved without any speckle reduction imaging or cross beam imaging post-processing to do the CTx analysis. Additionally, a video clip in which pressure was applied softly and progressively on the cervix until no further compression of the anteroposterior diameter could be observed. The technique used to apply the maximum compression was done as described by Parra-Saavedra et al. [11]. The images were downloaded from the medical imaging software and stored in a research imaging server for off-line analysis. Quality criteria to consider an image were: (1) the entire cervix seen and (2) the cervical canal not inclined more than 45° over the horizontal plane. All measurements were done blinded to the pregnancy outcome and measurement methods are depicted in Figure 1:
Standard CL (straight-line) measurement in mm [21].CCI measurement expressed as a percentage [11]. A straight cervical length was traced to align with the longitudinal axis of the cervix at the posterior vaginal wall. Then the midpoint of this line was calculated, and the anteroposterior diameter was measured perpendicular to this line with (AP’) and without (AP) pressure. Then the cervical consistency index was calculated with the following formula:
CCI = (AP′/AP) × 100(1)UCA calculated by measuring the angle between the straight anterior myometrial wall line and cervical length tracing [15].CTx was done as described using a machine learning algorithm [18].TVU cervical funnel analysis with and without fundal pressure and defined as protrusion of the amniotic membranes into the cervical canal. The longest and widest funnel length were taken at the internal os [21].TVU amniotic fluid sludge presence defined by the presence of hyperechogenic free floating particulate matter, in the proximity of the internal cervical os [21,22].

The endpoint was the ability to predict sPTB before 34^+0^ weeks, and the secondary endpoint was the ability to predict sPTB before 37^+0^ weeks, compared to that of sonographic CL.

Data distribution was assessed using the Shapiro–Wilk test of normality. The statistical significance of differences in continuous data was calculated using Student’s t-test or Mann–Whitney U-test for normally and non-normally distributed data, respectively, and in categorical data using the Chi-square test or Fisher’s exact test as appropriate. Group analysis of continuous variables with a normal distribution was performed using ANOVA for multiple comparisons. Simple liner regression determined the strength of association between the TVU markers and the GA at delivery as indicated by Pearson’s correlation coefficients. A stepwise logistic regression was used to identify correlates with sPTB before 34^+0^ weeks.

Receiver operating characteristic (ROC) curves with regard to predicting sPTB before 34^+0^ and before 37^+0^ weeks were drawn, and AUC with 95% CI was calculated. Sensitivity, specificity, positive and negative likelihood ratios (LR^+^ and LR^−^) with their 95% CI with regard to predicting sPTB < 37^+0^ and < 34^+0^ gestational weeks were calculated for the optimal cut-off point. The optimal cut-off is the one corresponding to that point on the ROC curve situated farthest from the reference line.

An a priori total sample size of 57 was calculated, of 38 controls and 19 cases. At the time of setting up the study, no twin CCI data was available, and the sample size was based upon a previous publication of CCI in high-risk singletons [10] that noted an AUC for CL of 0.51 and CCI of 0.73 and keeping account for the wide variety of AUCs published for the prediction of PTB in twins using CL [23,24]. This calculation included a type I error of 0.05, power of 0.90 with an allocation ratio of 2 and an anticipated sPTB rate of >20%.

## 3. Results

Initially, 121 women were screened, 9 declined transvaginal evaluation, and a total of 112 women underwent a TVU assessment at the time of the routine second-trimester scan. Of these, 63 women were included in the analysis. Flowchart summarizing the inclusion of women is presented in Figure 2.

Demographic details, cervical markers, and pregnancy outcomes according to gestational age at birth are shown in Table 1. All pregnancies were dated early between 74–91 days of gestation, and most women, 66.7% (42/63), used prophylactic low dose aspirin. The cohort had no PTB risk factors with regards to prior sPTB between 16 and 34 weeks, uterine abnormalities, or any conisation/cervical procedures. Conception by in vitro fertilization was noted in 13.9% (5/36) of the women delivering at term and was higher in the sPTB < 37^+0^ and < 34^+0^ groups, respectively 25.9% (7/27) and 31.3% (5/16), but this difference was not significant (*p* = 0.25). Nulliparity (*p* = 0.02) and a lower body mass index (BMI) at booking (*p* = 0.01) in the sPTB groups were the only cohort characteristic that was statistically different between the groups. Analyses could not be done in 5/63 UCA images, all due to retroflexion, and in 18/63 CTx images, mostly due to post-processing settings (16/18).

The rate of sPTB before 34^+0^ weeks was 16/63 (16.3%) and 27/63 (27.6%) delivered before 37^+0^ weeks. The CL (*p* = 0.01), CCI (*p* < 0.01), UCA (*p* < 0.01) but not the CTx, funnelling, or sludge were significantly different between term and sPTB before < 34^+0^ and < 37^+0^ weeks. The distribution and correlation plots of the ultrasound derived markers are shown in Appendix A and their correlation with GA in Appendix A. Multivariate logistic regression of the four novel ultrasound markers showed that only CCI (*p* = 0.03; OR 0.15 (0.07–0.35)), and UCA (*p* < 0.01; OR 1.08 (1.02–1.14)) were independently associated with sPTB before 34^+0^ weeks. Sensitivity, specificity as well as positive and negative likelihood ratios for the prediction of sPTB before 34^+0^ and 37^+0^ are given in Table 2 and Table 3, respectively.

CL had a low sensitivity even if the most optimal cut off for this cohort was chosen, while for the prediction of both sPTB before 34^+0^ and 37^+0^, the CCI had the best performance. Specifically, the AUC for sPTB < 34^+0^ prediction was 0.57 (95% CI, 0.34–0.79) for the CL, 0.82 (95% CI, 0.72–0.92) for the CCI, 0.72 (95% CI, 0.57–0.87) for the UCA, and 0.63 (95% CI, 0.47–0.79) for the CTx. For the prediction of sPTB < 37^+0^, the AUC was 0.63 (95% CI, 0.49–0.77) for the CL, 0.82 (95% CI, 0.72–0.92) for the CCI, 0.76 (95% CI, 0.63–0.88) for the UCA, and 0.63 (95% CI, 0.47–0.79) for the CTx (Figure 3).

A logistic regression model resulted in an AUC of 0.91 with a sensitivity of 55.3% and specificity of 88.1% for predicting sPTB before 34^+0^ if parity, chorionicity, CCI, and UCA were used.

## 4. Discussion

This study indicates that CCI is superior to other TVU markers for the prediction of sPTB < 34^+0^ (AUC 0.82; 95% CI, 0.72–0.92). Secondly, both CCI and UCA were independently associated with sPTB. When combining parity, chorionicity, CCI and UCA an AUC of 91.6% was reached.

Our study is the first to assesses the CCI, UCA, and CTx concurrently in a single prospective twin cohort using a standardized TVU approach. There are two main limitations of this study. All analysis was done off-line, which can lead to overestimating its performance. When analyses of images and measurements are done in real-time, there will be time and technical constraints. Moreover, several images had to be excluded due to insufficient quality criteria, especially in cases where there was retroflexion of the uterus. Consequently, these measurements cannot be performed in all women, as previously reported for CCI [10,25].

We observed a linear relationship between the GA at delivery and the CCI (R^2^ 0.3444), which is in agreement with the initial publication from Parra-Saavedra et al. [26]. Moreover, others have shown that the CCI positively correlates with the cervical elastography strain ratios, albeit only in singletons [27]. Ultimately the CCI was the strongest independent marker for sPTB < 34^+0^ and < 37^+0^ in this study. This stands in contrast to the only other publication that noted a limited role of a single mid-trimester CCI measurement in the prediction of sPTB (AUC 0.538) [28]. The value of the CCI has already been reported in low- [11,25] and high-risk [10] singletons. In these studies, it has been shown to be a reproducible and reliable sPTB predictor. The key concern is that the CCI is difficult to measure in real-time and that the force exerted by the operator is unstandardized. The second concern has already been addressed under experimental conditions where it has been shown that the change in force applied did not result in significant strain differences, implying that the CCI should be reproducible and robust in a clinical setting [12]. Real-time measurement could easily be achieved with a pre-programmed ratio calculator, and therefore, future studies should explore this marker as one variable in a predictive model.

Another biomechanical sPTB marker that has shown promise is the UCA [15]. In this cohort, the UCA was another strong independent marker of sPTB with good discriminatory value for sPTB < 34^+0^ weeks. This is in line with a previous retrospective twin study [17]. The UCA unfortunately also has some limitations, as with the CCI, namely that in a uterus with retroflexion it is not possible to do a measurement. Interestingly, in our cohort, none of the women where the measurement could not be done due to retroflexion delivered before < 34^+0^. It would be worthwhile to reproduce this observation in a larger study. The benefit of combining CCI and UCA is that if one of them could not be done due to technical limitations, the other was possible.

In singletons, there is a clear linear correlation between the CL and GA at delivery [29]. CL is a widely accepted marker for the prediction of sPTB especially in high risk pregnancies [30]. Our data supports previous findings in twin pregnancies that there is also a linear correlation between CL and GA, and therefore could be utilized as a sPTB marker [23]. Nevertheless, similar to recent reports, the prediction accuracy in our twin cohort remains limited, mainly due to low sensitivity [31]. Probably, CL shortening is most likely a “late” gross structural finding whilst the CCI reflect an earlier change in cervical microstructure and water concentration [32].

Consequently, it is sensible to search for other markers of early microstructural cervical changes. Quantitative ultrasonography of B-mode images has recently been introduced to indirectly evaluate the cervical collagen microstructure through speckle texture characterization and greyscale analysis. Quantitate CTx analysis has already been shown to be predictive of sPTB < 37^+0^ in singletons [18]. A preliminary twin study noted a 69% sensitivity and 70% specificity for the prediction of sPTB [19]. Our data could not confirm this. Possible explanations may be that the algorithm used was created on a singleton population hence not applicable to our twin cohort. Alternatively, the results were influenced by the high number of excluded images. Future studies should focus on reproducibility and easier acquisition protocol and then applying optimal algorithms before this could be seen as a potential tool to use in sPTB prediction.

We also made some unexpected observations. In this cohort, a lower BMI was more frequently seen in the sPTB groups, but ultimately, the BMI in all groups was within the normal range. Multiparity was seen more frequently in the sPTB groups. However, none had a previous history of sPTB. Funnelling was only observed in three women, and they all delivered < 37^+0^, two of them at < 34^+0^. The presence of sludge was only seen in two women. Lastly, selective intrauterine growth restriction (sIUGR) was much more prevalent in the sPTB < 34^+0^ group. These factors could potentially be used in future prediction models in combination with the CCI and UCA.

The strengths of this study include its prospective nature and the stringent quality criteria used for image analysis. The multimodal approach of the biomechanical cervical assessment allows for a more comprehensive “non-invasive” overview of the cervix. This was done through TVU, which is widely used and accepted by most women. This study adds to the emerging data that the CCI, and potentially the UCA, can have a role in sPTB prediction in twins.

Next to earlier named limitations, we acknowledge that this is only a small and single center study. Our population included mainly “low-risk” Caucasian women, and the novel methods used may be more difficult to implement reliably and in a broader setting.

The mechanism of sPTB in a multiple pregnancy is likely to be different from that in singletons. The heterogeneous nature of sPTB will need the incorporation of multiple markers or risk factors to correctly identify the patients at risk and the subsequent correct preventative intervention. For instance, maybe a structurally very short and “soft” cervix could benefit from a cerclage, while a long and “soft” cervix could benefit more from progesterone, and those with a long and extended cervix could maybe benefit from a pessary.

## 5. Conclusions

Mid-trimester CCI performed better than other TVU-derived biomechanical markers to predict sPTB < 34^+0^ in twin gestations, but its biggest shortcoming is that it is not yet a readily implementable clinical tool. Furthermore, in view of the moderate sensitivity and specificity of all these markers, this study highlights that a single parameter will not be enough to predict sPTB in twins accurately. The discriminative capacity of different models using pre-programmed measurements should be validated in a larger study. Lastly, future research should focus on the technical constraints of these novel markers so that they could be incorporated into daily practice.

## Figures and Tables

**Figure 1 jcm-09-01784-f001:**
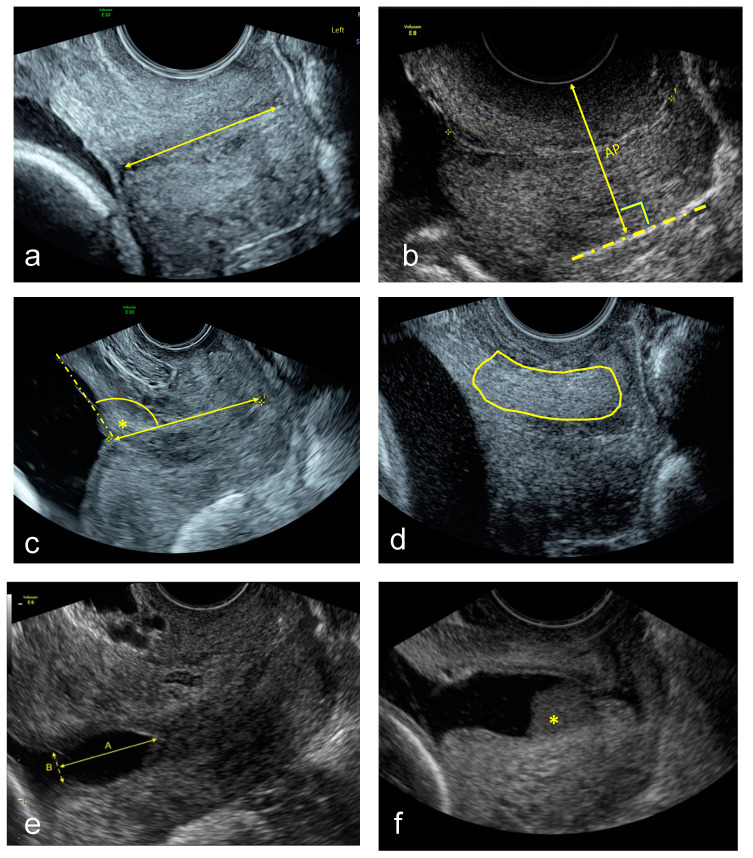
Example images of (**a**) Transvaginal ultrasound cervical length (straight-line) in mm. (**b**) Cervical consistency index measured as the ratio of the anteroposterior diameter with and without pressure and expressed as a percentage, CCI = (AP′/AP) × 100. (**c**) Utero-cervical angle calculated in degrees. (**d**) Cervical texture analysis. (**e**) Cervical funnel analysis, the longest and widest funnel at the internal os in mm. (**f**) Presence of amniotic fluid sludge (*).

**Figure 2 jcm-09-01784-f002:**
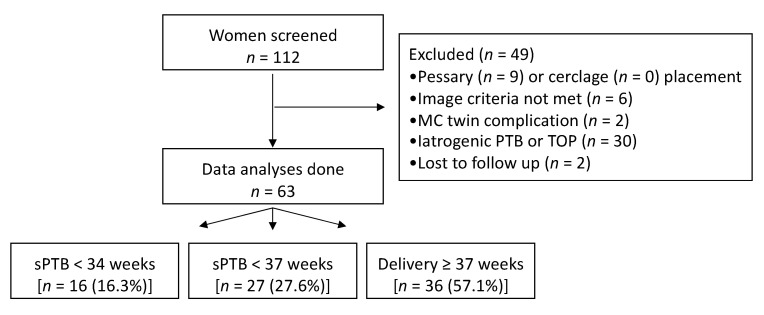
Flowchart summarizing the recruitment of women. Abbreviations: sPTB, spontaneous preterm birth; MC, monochorionic; TOP, termination of pregnancy.

**Figure 3 jcm-09-01784-f003:**
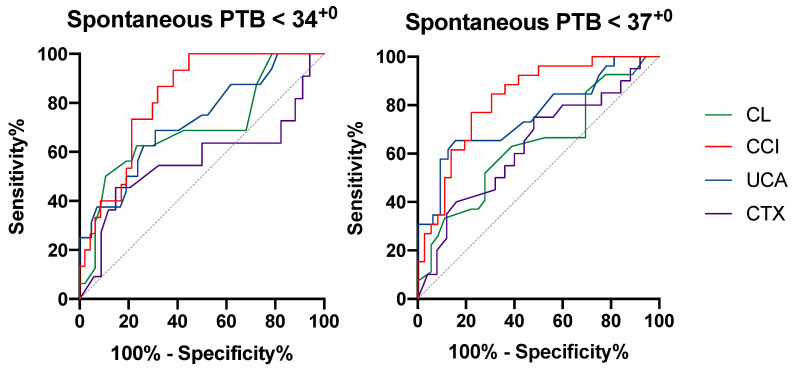
Receiver operating characteristic (ROC) curves for the cervical length (CL), cervical consistency index (CCI), utero-cervical angle (UCA), and cervical texture (CTx) and with regard to predicting spontaneous preterm birth (sPTB) before 34^+0^ and before 37^+0^ weeks.

**Table 1 jcm-09-01784-t001:** Demographic and pregnancy characteristics.

	Total *n* = 63	TB *n* = 36	sPTB < 34 w *n* = 16	sPTB < 37 w *n* = 27	*p*-Value ^†^
Maternal Age (Years)	31.2 ± 4.9	31.5 ± 5.2	29.9 ± 2.8	31.4 ± 5.6	*p* = 0.79
Caucasian Ethnicity	95.2% (60/63)	94.4% (34/36)	93.8% (15/16)	96.3% (26/27)	*p* = 0.92
BMI (kg/m^2^)	23.7 ± 4.9	24.6 ± 5.8	22.3 ± 2.4	22.4 ± 2.4	*p* = 0.01
Smoker	7.9% (5/63)	8.3% (3/36)	6.3% (1/16)	7.4% (2/27)	*p* = 0.95
Spont. Conception	57.1% (36/63)	77.8% (28/36)	62.5% (10/16)	66.7% (18/27)	*p* = 0.57
Nulliparous	31.7% (20/63)	30.8% (8/36)	62.5% (10/16)	44.4% (12/27)	*p* = 0.02
Dichorionicity	63.5% (40/63)	72.2% (26/36)	43.8% (7/16)	51.9% (14/27)	*p* = 0.09
GA at Scan (Days)	138 ± 17	135 ± 24	140 ± 4	141 ± 5	*p* = 0.48
Cervical Length (mm)	35.9 ± 5.7	37.3 ± 5.1	32.5 ± 6.6	34.2 ± 6.2	*p* = 0.02
CCI (%)	69.6 ± 10.3	74.7 ± 8.6	60.8 ± 8.0	63.1 ± 8.5	*p* < 0.01
UCA (°)	105.7 ± 16.8	98.5 ± 10.8	116.8 ± 20.3	114.7 ± 18.8	*p* < 0.01
CTx Score	−7.8 ± 3.2	−6.9 ± 3.2	−10.6 ± 3.6	−8.9 ± 2.8	*p* = 0.09
Funnelling (%)	47.6% (3/63)		12.5% (2/16)	11.1% (3/27)	*p* = 0.10
Sludge (%)	3.2% (2/63)	2.8% (1/36)	6.3% (1/16)	3.7% (1/27)	*p* = 0.83
Antenatal Complications					
GHT/Preeclampsia	1.6% (1/63)	2.8% (1/36)			
GDM	3.2% (2/63)	5.6% (2/36)			
sIUGR	4.8% (3/63)	2.8% (1/36)	12.5% (2/16)	7.4% (2/27)	*p* = 0.39
Received Progesterone	4.8% (3/63)	2.8% (1/36)	6.3% (1/16)	7.4% (2/27)	*p* = 0.68
Received Betamethasone	39.7% (25/63)	16.7% (6/36)	93.8% (15/16)	70.4% (19/27)	*p* < 0.01
Delivery Outcomes					
GA at Delivery (w d)	35 w 2 d ± 25 d	37 w 3 d ± 4 d	30 w 1 d ± 24 d	32 w 4 d ± 27 d	*p* < 0.01
Birth Weight-Largest (g)	2312 ± 623	2683 ± 298	1490 ± 459	1882 ± 629	*p* < 0.01
Birth Weight-Smallest (g)	2131 ± 616	2506 ± 270	1333 ± 463	1711 ± 627	*p* < 0.01
Spontaneous Labor	58.7% (37/63)	27.8% (10/36)	100% (16/16)	100% (27/27)	*p* < 0.01
Induced Labor	20.6% (13/63)	36.1% (13/36)			
Vaginal Delivery	52.4% (33/63)	50.0% (18/36)	50.0% (8/16)	55.5% (15/27)	*p* = 0.89
Elective Caesarean	19.0% (12/63)	33.3% (12/36)			
Emergency Caesarean	28.6% (18/63)	16.7% (6/36)	50.0% (8/16)	44.4% (12/27)	*p* = 0.02

Data are given as mean ± SD or *n* (%). In UCA 5/63 and CTx 18/63 could not be analysed; ^†^ two-way ANOVA or Chi-square test. Abbreviations: TB, term birth; sPTB, spontaneous preterm birth; BMI, body mass index; GA, gestational age; CL, cervical length; CCI, cervical consistency index; UCA, uterocervical angel; CTx, cervical texture score; GHT, gestational hypertension; GDM, gestational Diabetes Mellitus; sIUGR, selective intrauterine growth restriction; SD, standard deviation.

**Table 2 jcm-09-01784-t002:** Discriminative performance of the UCL, CCI, UCA, and CTx measurements with regard to predicting spontaneous preterm birth < 34^+0^ weeks, with optimal cut-off based on the receiver operating characteristic curve given.

	Cut-Off	Sensitivity (95% CI)	Specificity (95% CI)	LR ^+^ (95% CI)	LR ^−^ (95%CI)	dOR (95% CI)
CL	<29 mm (*p* = 10)	31.3 (11.0–58.7)	93.6 (82.5–98.7)	4.90 (1.32–18.22)	0.73 (0.52–1.03)	6.7 (1.37–32.25)
	<35 mm	62.5 (35.4–84.8)	70.2 (55.1–82.7)	2.10 (1.17–3.75)	0.53 (0.28–1.03)	3.9 (1.19–12.9)
CCI	<60% (*p* = 10)	40.0 (16.3–67.7)	91.5 (79.6–97.6)	4.70 (1.53–14.46)	0.66 (0.43–1.00)	7.2 (1.67–30.78)
	<64%	86.7 (59.5–98.3)	65.9 (50.7–79.1)	2.055 (1.63–3.97)	0.20 (0.05–0.75)	12.6 (2.52–62.77)
UCA	>130° (*p* = 10)	40.0 (16.3–67.7)	91.5 (79.6–97.6)	4.70 (1.53–14.46)	0.66 (0.43–1.00)	6.7 (1.06–25.92)
	>103°	68.8 (41.3–88.9)	73.8 (57.9–86.1)	2.63 (1.43–4.81)	0.42 (0.20–0.90)	6.2(1.76–21.89)
CTx	−13.5 (*p* = 10)	27.3 (6.0–60.9)	88.2 (72.5–96.7)	2.32 (0.61–8.80)	0.82 (0.56–1.21)	2.8 (0.52–15.2)
	−11.0	72.7 (39.0–93.9)	38.2 (22.2–56.4)	1.18 (0.75–1.84)	0.71 (0.25–2.05)	1.65 (0.37–7.37)

Abbreviations: CL, cervical length; CCI, cervical consistency index; UCA, utero-cervical angle; CTx, cervical texture; LR, likelihood ratio; CI, confidence interval; dOR, diagnostic odds ratio. +, positive; −, negative.

**Table 3 jcm-09-01784-t003:** Discriminative performance of the CL, CCI, UCA, and CTx measurements with regard to predicting spontaneous preterm birth < 37^+0^ weeks, with optimal cut-off based on the receiver operating characteristic curve given.

	Cut-Off	Sensitivity (95% CI)	Specificity (95% CI)	LR ^+^ (95% CI)	LR ^−^ (95%CI)	dOR (95% CI)
CL	<29 mm (*p* = 10)	22.2 (8.6–42.3)	94.4 (81.3–99.3)	4.00 (0.87–18.30)	0.82 (0.66–1.02)	4.9 (0.89–26.33)
	<36 mm	51.8 (31.9–71.3)	72.2 (54.8–85.8)	1.87 (0.98–3.54)	0.67 (0.43–1.04)	2.8 (0.98–7.99)
CCI	<60% (*p* = 10)	30.8 (14.3–51.8)	94.4 (85.5–99.9)	5.54 (1.28–23.97)	0.73 (0.56–0.96)	7.6 (1.45–39.40)
	<68%	76.9 (56.3–91.0)	75.0 (57.8–87.8)	3.08 (1.68–5.63)	0.31 (0.15–0.64)	10.0 (3.06–32.67)
UCA	>130° (*p* = 10)	23.1 (8.9–43.7)	100 (89.1–100)		0.77 (0.62–0.95)	
	>105°	65.4 (44.3–82.8)	84.4 (67.2–94.7)	4.18 (1.78–9.81)	0.41 (0.24–0.71)	10.2(2.92–35.61)
CTx	−11.3 (*p* = 10)	20.0 (5.73–43.7)	96.0 (79.7–99.9)	5.00 (0.61–41.28)	0.83 (0.66–1.05)	6.0 (0.61–58.71)
	−8.5	60.0 (36.1–80.9)	64.0 (42.5–82.0)	1.67 (0.88–3.14)	0.62 (0.34–1.15)	2.7 (0.79–8.95)

Abbreviations: CL, cervical length; CCI, cervical consistency index; UCA, utero-cervical angle; CTx, cervical texture; LR, likelihood ratio; CI, confidence interval; dOR, diagnostic odds ratio. +, positive; −, negative.

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
