# Peer review of "The Predictive Value of the Cervical Consistency Index to Predict Spontaneous Preterm Birth in Asymptomatic Twin Pregnancies at the Second-Trimester Ultrasound Scan: A Prospective Cohort Study"

_jcm, 2020, doi:10.3390/jcm9061784_

Round 1

Reviewer 1 Report

Very interesting study and well written manuscript. There are only some minor remarks. In table 1 days are shown for the gestational age at delivery, I would recommend the use of weeks. In the contribution section is a typo: All authors have contributed tis his paper.

Author Response

Reviewer #1:

  1. “In table 1 days are shown for the gestational age at delivery, I would recommend the use of weeks.” I have converted the days to reflect weeks and days. See table 1.
  2. “In the contribution section is a typo: All authors have contributed tis his paper.” Thank you for spotting this mistake. Typo corrected line 293.

Reviewer 2 Report

The authors reported a prospective cohort study of asymptomatic twin pregnancies with transvaginal ultrasound assessment between 18+0 and 22+0 weeks. They demonstrated that cervical consistency index is a good marker to predict sPTB < 34+0 in twin gestations.

The topic is very interesting because some markers, including ultrasound markers, have been evaluated in twins with limited success, and literature is poor about proved intervention to predict and prevent sPTB in twins.

The manuscript is very well-written, with minor details should be modified to increase the interest for the physicians. The methodology is close to a proof of concept study with a small sample size (n=63), a monocentric study, low-risk Caucasian women, and an evaluation of many ultrasound markers to highlight the most interesting one.

In details: CCI measurement should be more detailed with explanations noted to the Figure 1.

The sample size calculation should use references about twins and not about high risk singleton pregnancies.

CCI that is the best marker is operator-dependent and difficult to measure in real-time (clearly noted by the authors as a great limitation), and authors should then moderate conclusions.

The authors should also moderate results and conclusions because only AUC of at least 0.80 was considered to represent accurate prediction [Hanley 1982, McNeil 1984].

Some references should be rewritten to ensure the uniformity of the manuscript.

Author Response

Reviewer #2: 

  1. “CCI measurement should be more detailed with explanations noted to the Figure 1.” Figure 1 and the legend of Figure 1 have been amended. Line 115-117.
  2. “The sample size calculation should use references about twins and not about high risk singleton pregnancies.” At the time of setting up the study no twin CCI data existed and the “most appropriate” data we could use was the description of a high-risk singleton cohort. Explanation added to line 139-140.
  3. “CCI that is the best marker is operator-dependent and difficult to measure in real-time (clearly noted by the authors as a great limitation), and authors should then moderate conclusions. The authors should also moderate results and conclusions because only AUC of at least 0.80 was considered to represent accurate prediction [Hanley 1982, McNeil 1984].” Absolutely correct, the comparison in this study was made between the different markers. Although the CCI did reach an UAC of 0.82 and the regression model an AUC of 0.91 for prediction sPTB <34w, we fully acknowledge all the shortcomings of this study. The conclusions have been rewritten, line 280-284.
  4. “Some references should be rewritten to ensure the uniformity of the manuscript.” We used endnote as a citation manager but correctly spotted there were some inconsistencies. I have amended reference 2, 27. 
